# Sequencing of Historical Isolates, K-mer Mining and High Serological Cross-Reactivity with Ross River Virus Argue against the Presence of Getah Virus in Australia

**DOI:** 10.3390/pathogens9100848

**Published:** 2020-10-16

**Authors:** Daniel J. Rawle, Wilson Nguyen, Troy Dumenil, Rhys Parry, David Warrilow, Bing Tang, Thuy T. Le, Andrii Slonchak, Alexander A. Khromykh, Viviana P. Lutzky, Kexin Yan, Andreas Suhrbier

**Affiliations:** 1Inflammation Biology Group, QIMR Berghofer Medical Research Institute, Brisbane, QLD 4006, Australia; Daniel.Rawle@qimrberghofer.edu.au (D.J.R.); Wilson.Nguyen@qimrberghofer.edu.au (W.N.); Troy.Dumenil@qimrberghofer.edu.au (T.D.); Bing.Tang@qimrberghofer.edu.au (B.T.); Thuy.Le@qimrberghofer.edu.au (T.T.L.); Viviana.Lutzky@qimrberghofer.edu.au (V.P.L.); Kexin.Yan@qimrberghofer.edu.au (K.Y.); 2School of Chemistry and Molecular Biosciences, University of Queensland, Brisbane, QLD 4072, Australia; r.parry@uq.edu.au (R.P.); a.slonchak@uq.edu.au (A.S.); alexander.khromykh@uq.edu.au (A.A.K.); 3Public Health Virology Laboratory, Department of Health, Queensland Government, Brisbane, QLD 4108, Australia; David.Warrilow@health.qld.gov.au; 4GVN Center of Excellence, Australian Infectious Diseases Research Centre, Brisbane, QLD 4006 and 4072, Australia

**Keywords:** Getah virus, Ross River virus, Sequence Read Archive, serology, mouse model, virus contamination

## Abstract

Getah virus (GETV) is a mosquito-transmitted alphavirus primarily associated with disease in horses and pigs in Asia. GETV was also reported to have been isolated from mosquitoes in Australia in 1961; however, retrieval and sequencing of the original isolates (N544 and N554), illustrated that these viruses were virtually identical to the 1955 GETV_MM2021_ isolate from Malaysia. K-mer mining of the >40,000 terabases of sequence data in the Sequence Read Archive followed by BLASTn confirmation identified multiple GETV sequences in biosamples from Asia (often as contaminants), but not in biosamples from Australia. In contrast, sequence reads aligning to the Australian Ross River virus (RRV) were readily identified in Australian biosamples. To explore the serological relationship between GETV and other alphaviruses, an adult wild-type mouse model of GETV was established. High levels of cross-reactivity and cross-protection were evident for convalescent sera from mice infected with GETV or RRV, highlighting the difficulties associated with the interpretation of early serosurveys reporting GETV antibodies in Australian cattle and pigs. The evidence that GETV circulates in Australia is thus not compelling.

## 1. Introduction

The mosquito-transmitted, arthritogenic alphaviruses that cause rheumatic disease in humans include chikungunya virus (CHIKV), Ross River virus (RRV), Barmah Forest virus, Sindbis virus, o’nyong’nyong virus (ONNV) and Mayaro virus (MAYV) [1]. CHIKV was responsible for the recent global pandemic that started in Africa in 2004, spread to >100 countries on four continents and involved >10 million cases [2]. The Australasian RRV is endemic in Australia with ≈4000 cases annually [3]. RRV also caused an outbreak of >60,000 cases in the Pacific Islands in 1979/80 [4], and arguably has the potential for global spread [5]. Another member of this family of alphaviruses is Getah virus (GETV), which causes disease primarily in horses and pigs, with evidence for human infections limited to a small number of early serology studies undertaken before the advent of ELISA-based technologies [6,7]. GETV belongs to the Semliki Forest virus antigenic complex, that includes RRV, CHIKV, MAYV and ONNV, and is most closely related to RRV [8,9].

GETV was first isolated in Malaysia from *Culex gelidus* mosquitoes in 1955 and has since been isolated from mosquitoes, pigs, foxes and horses across several Asian countries [10,11,12,13,14], with GETV also recently isolated from a horse in China [12]. Outbreaks of GETV in racehorses in Japan [11,15] prompted the production and use of a commercial (Nisseiken) formalin-inactivated, two shot, whole-virus vaccine that is sold as a mixed, two-component vaccine comprising Japanese Encephalitis virus and GETV (JE/GETV) [15,16,17]. Clinical signs of GETV infection in horses include fever, swelling of the hind limbs and lymph nodes, and rash [18,19]. GETV also represents a potential emerging burden for the pig industry, recently highlighted for China [20]. Infected piglets show depression, tremors, hind limb paralysis, diarrhea and high mortality, and infection of sows can be associated with abortion [21]. Live GETV was recently isolated as an adventitious agent contaminating a commercial live-attenuated vaccine for porcine reproductive and respiratory syndrome virus (manufactured in MARC-145 cells) [22]; perhaps a contributing factor in the increased GETV burden in pigs. GETV was also recently isolated from a cow presenting with a fever in north-eastern China; the first GETV isolation from cattle [23].

GETV was also reported to be present in Australia, with two isolates N544 and N554 obtained from *Anopheles amictus amicus* and *Culex bitaeniorhynchus* mosquitoes, respectively, in Normanton in Northeastern Queensland, Australia in 1961 [24]. These virus isolates were grouped by serology with MM2021 [24], a virus later classified as GETV [25]. Early sero-surveillance also identified GETV-reactive antibodies in cattle [26] and pigs [27] in Australia using hemagglutination inhibition (HI) assays [28] to measure anti-viral antibody levels and lethal dose titrations in suckling mice to assess neutralizing antibody levels [29]. Anti-RRV antibodies have also been detected in cattle and pigs [26,27,30,31,32], with these species also able to be infected experimentally with RRV [33]. Early serosurveys [26,27] frequently found sera that reacted with both RRV and GETV. Nevertheless, in these studies (i) 22 out of 1389 serum samples from cattle reacted with GETV (but not RRV) by HI, with 4 showing anti-GETV neutralizing activity [26] and (ii) out of 239 pig sera, 4 reacted with GETV (but not RRV) by HI and 1 of these neutralized only GETV [27]. Together with isolation of N554 and N544, these serosurveys [26,27] have led to the established view that GETV circulates in Australia [31,34,35,36], with multiple papers and reviews including Australia in the list of countries where GETV circulates [37,38,39].

Herein, we recover N554 from an original vial frozen in 1961 and N544 (frozen in 1983) and show that that they are virtually identical to the GETV_MM2021_ virus isolated in Malaysia from *Culex gelidus* mosquitoes in 1955 [8]. Bioinformatic interrogation of the >40,000 terabases of open access sequence data publicly available via the Sequence Read Archive (SRA), also failed to identify GETV sequences in biosamples collected in Australia. To further investigate the serological cross-reactivity between GETV and RRV, we established an adult wild-type mouse model of GETV using GETV_MM2021_. A high level of overlapping cross-reactivity and cross-neutralization between convalescent RRV and GETV sera became apparent, perhaps arguing that early Australian GETV serosurveys [6,7,26,27] may have not have been reliably able to distinguish between past infections with GETV and past infections with RRV. Thus, despite an abundance of pigs, horses, cattle and foxes in Australia, and mosquito species likely capable of transmitting GETV, evidence that GETV circulates in Australia would thus appear not to be overly compelling.

## 2. Results

### 2.1. Sequencing of the Purported Australian GETV Isolates, N544 and N554

GETV has been isolated in several Asian countries (Figure 1). The contention that GETV also circulates in Australia [26,27,31,34,35,36] is based primarily on the isolation of N544 and N554 from mosquitoes trapped in Australia in 1961 [24]. Both N544 and N554 were stored in the “Doherty Virus Collection”, recently relocated to the QIMR Berghofer Medical Research Institute. An original vial of N554, frozen in 1961 (Appendix A), and a vial of N544 (frozen in 1983) were thawed, cultured in C6/36 cells and sequenced. The collection also contained the 1955 Malaysian GETV_MM2021_ isolate (frozen in 1986), for which we previously provided the complete genome sequence [8] (Genbank ID: MN849355). Partial sequence for GETV_MM2021_ uploaded independently (Genbank ID: AF339484) showed 99.69% percent nucleotide identity with our GETV_MM2021_ sequence (Genbank ID: MN849355).

The sequences of N544 and N554 were found to be nearly identical to each other and to GETV_MM2021_ (Figure 1 and Appendix A). The small differences are likely due to different passage histories with file notes associated with the “Doherty Virus Collection” indicating that N554 was frozen in 1961 after passage in mouse brains, N544 in 1983 after 5 passages in vitro and GETV_MM2021_ in 1986 after passage in mouse brains. Two very different sequencing protocols were used for N544 and N554 (see Section 4), with culture and sequencing undertaken ≈4 months apart arguing against any form of cross-contamination. These data argue that N544 and N554 are not unique Australian GETV isolates, but likely arose from contamination with GETV_MM2021_, given GETV_MM2021_ was present in the laboratory at the time of purported N544 and N554 isolation [24].

### 2.2. GETV K-mer Mining and BLASTn Confirmation of High Throughput Sequencing Data

To the best of our knowledge, no other sequence-based evidence for the presence of GETV in Australia has been reported. To determine whether any GETV sequences might have been deposited in high throughput sequencing data, the Google BigQuery service offered by Google Cloud Computing Service was used to search for GETV k-mers (short nucleotide sequences constructed from the entire GETV genome) in the Sequence Read Archive (SRA), hosted by the National Center for Biotechnology Information. SRA submissions that were found to contain GETV k-mers were then separately subjected to BLASTn searches to confirm the presence of GETV reads in the high throughput sequencing data using the entire GETV genome as an input query. SRA submissions where BLASTn confirmed the presence of GETV reads (Appendix A) are shown in Table 1. All biosamples containing GETV sequences originated from Asian countries where GETV has previously been isolated and sequenced (Table 1, Figure 1). Although one SRA originated from Sweden, the biosample was obtained from China (Table 1; Bioproject PRJNA596441). Confirmed GETV reads were found in mosquito sequencing SRA submissions from China (Table 1, Bioproject PRJNA271540), whereas they were not found in wild-caught mosquitos from Victoria or Queensland, Australia (Table 2). Other studies of mosquitoes from Western Australia reported not finding any arbovirus sequences [41] or sequences with homology to known human or mammalian pathogens [42].

BLASTn alignment of RNA-Seq data from an experiment using the mouse embryonic fibroblast NIH 3T3 cell line (Table 1, Bioproject PRJNA561663) provided almost complete genome coverage of GETV (Appendix A). Curiously, k-mer mining and BLASTn confirmation of SRA submissions from Whole Genome Sequencing (WGS) of DNA samples revealed samples with GETV sequences (Table 1). GETV sequences were also found in a series of plant (apple, soy, pomegranate, maize, and artichoke), fish and bacteria samples; such biosamples would not be expected to contain GETV. These GETV sequences likely arise from unknown contamination events (see Section 3). 

This interrogation of >40,000 terabases of open access sequence data thus provided ample evidence for the presence of GETV in Asia but failed to provide any evidence for the presence of GETV in Australia.

### 2.3. RRV K-mer Mining and BLASTn Confirmation of High Throughput Sequencing Data

As a positive control for GETV k-mer mining, using the same approach used above for GETV, we conducted RRV k-mer mining of high throughput sequencing data followed by BLASTn confirmation. Multiple biosamples collected in Australia contained RRV k-mers, with BLASTn confirming the presence of RRV reads in the indicated SRA submissions (Table 2, Appendix A). Confirmed RRV reads were identified in sequence data sets derived from a mixed pool of wild-caught mosquitoes from Victoria (Australia) (BioProject: PRJNA343688) [43], five libraries of bulk wild-caught mosquito also from Victoria (BioProject: PRJNA642916), and *Aedes vigilax* collected at Shoal Water Bay Defense Training Area (Queensland, Australia) (Bioproject: PRJNA615690). Not surprisingly, experiments in Australian laboratories involving RNA-Seq of *Culex australicus* spiked with RRV (Bioproject; PRJNA559742) [44], *Aedes nostocriptus* infected with RRV (Bioproject; PRJNA386415) [45], and genome sequencing of RRV isolates (Bioproject; PRJNA522026) [46], all contained abundant RRV reads covering the whole genome.

As for GETV, RRV reads were also identified in Whole-genome DNA sequencing studies (Table 2, WGS), again suggesting contamination. Two of these were submitted by Queensland based laboratories (Table 2, Bioprojects: PRJNA606985, PRJNA639216), with Queensland the state with the highest number of RRV cases in Australia [47]. However, one was a malaria sample collected from the China/Myanmar border, with whole-genome sequencing submitted by a US institution (Table 2, Accession SRR8291079) that also works on alphaviruses. Approximately 0.01% of reads in this submission mapped to RRV, with similar read counts for structural and non-structural genes (Appendix A). The closest match being the prototype RRV T48 strain [8] (97.63% identity), with mismatches mostly uncalled nucleotides (Appendix A). These results perhaps argue against replicating or circulating RRV in China.

K-mer mining and BLASTn confirmation were, therefore, able to identify RRV sequences in multiple SRA datasets from Australia. As RRV is known to circulate in Australia, these results illustrate the ability of the k-mer mining and BLASTn confirmation approach to identify alphaviruses in those countries where they are known to circulate.

### 2.4. An Adult-Wild-Type Mouse Model of GETV Infection and Disease

To analyze cross-reactivity between GETV and RRV antisera (see below), we first established and partially characterized an adult-wild type mouse model of GETV infection and disease. Previous mouse models used young [48,49] or pregnant mice [36,50], illustrating that mice can be infected with GETV.

Infection of adult female C57BL/6J mice s.c. into the hind feet with GETV_MM2021_ resulted in a short viremia, cleared by day 4 post-infection (Figure 2A). H&E of feet day 6 post-infection showed inflammatory infiltrates in the muscle (myositis) and around tendons (tendonitis), with mild edema and occasional hemorrhage, all features seen previously in mouse models of alphaviral arthritides (Figure 2B) [51,52,53,54,55,56]. Mice were vaccinated twice with 20 µg of the JE/GETV vaccine (Nisseiken) [8], which contains the 1978 Japanese GETV_MI-110_ horse isolate [11]. All mice produced antibodies detectable by GETV antigen ELISA (Figure 2C), and 4/6 mice showed neutralizing activity against GETV_MM2021_ (Figure 2D). All animals were completely protected against both a detectable viremia (Figure 2E) and foot swelling (Figure 2F) after GETV_MM2021_ challenge. Thus GETV_MM2021_ and C57BL/6J mice can be used to establish an adult mouse wild-type model of infection and disease that is broadly comparable with other models of alphaviral arthritides [8], although viremia is comparatively shorter and foot swelling is mild. The ability of the current JE/GETV vaccine to protect against challenge with the 1955 GETV_MM2021_ isolate also supports the view that the original and contemporary GETVs are serologically closely related. Neutralizing titers >1 in 3 were reported to be protective in mice challenged with RRV [57], so the low post-vaccination GETV neutralizing titers seen herein (Figure 2D, mean 13.7 ± SE 5.1) might be deemed sufficient for protection.

### 2.5. High Levels of Cross-Reactivity between GETV and RRV Antisera

That RRV and GETV antibodies can cross-react and cross-neutralize was reported in a series of early studies [6,7,26,27]. We also recently showed that the JE/GETV vaccine could (i) induce antibodies in mice that partially cross-reacted with RRV in ELISA assays and (ii) provide partial cross-protection against RRV challenge [8]. To further characterize this cross-reactivity using ELISA and cell culture-based micro-neutralization assays, mice were infected with GETV, RRV or CHIKV and convalescent sera (>1-month post-infection) used in whole-virus antigen ELISAs, with GETV or RRV as the ELISA antigen. When GETV was used as the ELISA antigen, convalescent serum from GETV, RRV and CHIKV infected mice all showed a positive result, well above the limit of detection (Figure 3A), consistent with the fact that all three viruses belong to the Semliki Forest serogroup. The same was true when RRV was used as the ELISA antigen (Figure 3B). In neutralization assays, RRV antiserum was consistently able to neutralize GETV (Figure 3C), and GETV antiserum samples were consistently able to neutralize RRV (Figure 3D). Neither RRV nor GETV antisera were able to neutralize CHIKV (Figure 3C,D), consistent with previous reports showing that a CHIKV vaccine was unable to induce antibodies capable of neutralizing RRV [8]. Importantly, in all these assays RRV and GETV antisera gave overlapping titers, illustrating that these assays could not reliably distinguish between past infection with GETV or past infection with RRV for any given individual.

### 2.6. Cross-Protection between GETV and RRV

To investigate cross-protection, mice were infected with GETV, RRV or CHIKV and after 4 weeks were challenged with GETV. RRV and CHIKV immune mice were protected against the development of a detectable GETV viremia after GETV challenged (Figure 3E). Similarly, GETV immune mice were protected against the development of a detectable RRV viremia after RRV challenge (Figure 3F). In contrast, GETV immune mice were only partially protected against CHIKV challenge (Figure 3G). Past CHIKV infection protected against GETV challenge (Figure 3E), whereas past GETV infection only partially protected against CHIKV challenge (Figure 3G). This might be explained by the “immunizing” viremia for CHIKV (3 days of viremia >4 log_10_CCID_50_/mL; Figure 3G, Naive) being much higher than the “immunizing” viremia for GETV (1 day of viremia >4 log_10_CCID_50_/mL; Figure 3E, Naive). An overall stronger immune response would likely result in higher levels of cross-protective immunity; phenomena reported previously in related arthritogenic alphavirus cross-protection studies [8]. Importantly, the levels of cross-protection between RRV and GETV would appear to be high, consistent with the high levels of cross-neutralization.

## 3. Discussion

The best evidence that GETV circulates in Australia came from two virus isolations [24]; however, the sequence evidence presented herein illustrates that the “Australian” GETV isolates were actually the Malaysian GETV_MM2021_ isolate. Several serosurveys also indicated the presence of circulating GETV in Australia [6,7,26,27]. However, the number of sera deemed to be GETV positive, rather than RRV positive, was very low and the techniques used have several issues. HI assays were often difficult to standardize [28] and measuring neutralization titers using mortality readouts in suckling mice [29] may be complicated by the different abilities of GETV and RRV to replicate in murine systems (see Figure 3E,F). Circulation of RRV causing human disease in Australia and the Pacific Islands is well described [1,3,4,58], with a range of animals species (including cows, horses and pigs) also infected [30,33,59,60,61,62]. The very high level of overlapping cross-reactivity illustrated herein between sera known to be raised by RRV or GETV infections, argues that it would be difficult to distinguish between past infections with GETV and past infections with RRV. The large range of titers encountered herein and in the field [26,27,31,33] further complicate attempts at distinguishing between past GETV and RRV infections. Even using modern techniques, a methodology that reliably excludes RRV cross-reactivity would need to be developed before GETV positive serology could be readily claimed in a geographic area where RRV circulates, and for an animal species known to be infected with RRV. Most current ELISA alphavirus diagnostic assay systems use a positive/negative scoring system, generally applying a minimum threshold for scoring a positive serology result [3]; such systems would clearly be unable to distinguish between past GETV and RRV infections. In summary, the current serological evidence that GETV circulates in Australia is not overly compelling, and such data would not be straightforward to generate. 

Perhaps the more novel evidence presented herein is the use of k-mer mining of the >40,000 terabases of publicly available sequence data in the SRA, followed by confirmation with BLASTn. The process identified GETV sequences in biosamples from Asia, but not in biosamples from Australia (Table 1). In contrast, RRV reads (but not GETV reads) were identified in several pools of wild-caught mosquitoes collected in Australia, with RRV well known to circulate in Australia [1,4,47,58]. Clearly, lack of evidence is not proof of absence, with more extensive sampling of mosquitos from around Australia (e.g., northern Queensland [26]) perhaps warranted to support the current findings. However, no GETV sequences have so far been found in Australian biosamples, whereas RRV sequences were readily identified. 

Intriguingly, several k-mer “hits” were obtained in SRA submissions of Whole Genome DNA sequencing and were confirmed using BLASTn (Table 1 and Table 2, WGS). Although reverse transcription of RNA virus genomes in mammalian systems has been reported in certain settings [63,64,65,66], this has not been widely confirmed or reported for alphaviruses. GETV and RRV reads were also identified in biosamples unlikely to contain replicating alphaviruses. Such reads thus likely arose as a result of contamination, with contamination of high throughput sequencing samples a well-recognized phenomenon [67,68]. The high sensitivity of current high throughput sequencing technology means that even low-level contamination can be detected. Aside from biosample contamination, infection of cell lines with adventitious viruses is also a well-documented phenomenon [69,70,71]. For instance, Phasi Charoen-like virus and Cell fusing agent virus can be found in the Aag2 mosquito cell line [72], a rhabdovirus infects Sf9 cells [73], and a bovine polyomavirus infection was identified in the breast cancer cell line, SK-BR-3 [74]. Contamination of MARC-145 cultures with GETV was recently reported [22]; and herein we illustrate GETV contamination of NIH 3T3 cultures (Table 1, Bioproject PRJNA561663), although it is unlikely that all MARC-145 and NIH 3T3 cell cultures globally are similarly contaminated. That unforeseen sequencing of contaminants might provide an extra tool for viral molecular epidemiology is a novel concept, but relies on the contention that viruses circulating in a given country only contaminate biosamples from that country. Although the contention is generally supported by the GETV data presented herein, the large volume of biological samples and reagents exchanged internationally may often mean that follow-up investigations are needed (e.g., Appendix A).

The obvious question might arise, why would GETV not circulate in Australia? GETV can be transmitted by several mosquito species, including *Aedes aegypti* [75], which is well established in Australia [72]. Conceivably, amplifying hosts (e.g., horses and pigs) are of insufficient population density and/or such animals have already been infected with RRV and are thus protected from GETV. Native Australian macropods, such as the kangaroo and wallaby, are believed to be the most important enzootic vertebrate hosts for RRV [76]. Such animals may not be effective amplifying hosts for GETV, with GETV showing a distinct propensity to infect certain ungulates (Figure 1). Perhaps noteworthy is that Australia has an estimated 24 million feral pigs [77] and 400,000 feral horses [78], with the potential for arthritogenic alphavirus outbreaks well described [1,2,5,47,79].

## 4. Materials and Methods

### 4.1. Animal Ethics Statement

All mouse work was conducted in accordance with the “Australian code for the care and use of animals for scientific purposes” as defined by the National Health and Medical Research Council of Australia. Mouse work was approved by the QIMR Berghofer Medical Research Institute Animal Ethics Committee (P2235, A1606-618M). Serum was collected via tail bleed and mice were euthanized using CO_2_.

### 4.2. Cell Culture

The African green monkey kidney, Vero cell line (ATCC#: CCL-81) was maintained in RPMI 1640 (Thermo Fisher Scientific, Scoresby, VIC, Australia) supplemented with endotoxin-free 10% heat-inactivated fetal bovine serum (FBS; Sigma-Aldrich, Castle Hill, NSW, Australia) at 37 °C and 5% CO_2_. The *Aedes albopictus* mosquito larva-derived cell line, C6/36 (ATCC# CRL-1660) was cultured in RPMI 1640 with 10% heat-inactivated FBS at 28 °C and 5% CO_2_. Cells were checked for mycoplasma using MycoAlert Mycoplasma Detection Kit (Lonza, Basel, Switzerland) and FBS was checked for endotoxin contamination before purchase [80].

### 4.3. GETV N554 and N544 Recovery and Sequencing

GETV N554 was grown in C6/36 cells from an original vial frozen in 1961 and stored in the “Doherty Virus Collection” which was recently returned to QIMR Berghofer Medical Research Institute, Brisbane, Australia. After 3 days of culture, viral RNA was isolated from culture supernatant using TRIzol reagent as per manufacturer’s instructions (Life Technologies). cDNA was synthesized from viral RNA using ProtoScript II Reverse Transcriptase (New England Biolabs). Barcoding PCR was performed with GETV_MM2021_-specific primers containing the Nanopore universal tail sequences of 5′-TTTCTGTTGGTGCTGATATTGC-3′ for the forward primer and 5′-ACTTGCCTGTCGCTCTATCTTC-3′ for the reverse primer. Primers were designed to amplify approximately 1 kb overlapping amplicons spanning the entire GETV genome (Appendix A). PCR using pooled primers (odd and even-numbered primers pooled separately in two reactions to avoid primer interference at the overlapping sequences) was performed using Q5 High-Fidelity 2X Master Mix (New England Biolabs), and amplicons were purified using QIAquick Gel Extraction Kit (QIAGEN). The two amplicon pools for each sample were further pooled and the second round of barcoding PCR was performed using LongAmp Taq 2X Master Mix (New England Biolabs) and Oxford Nanopore barcoding primer set BC01 containing a unique barcode. DNA repair and end-prep using NEBNext FFPE DNA Repair Mix and NEBNext Ultra™ II End Repair/dA-Tailing Module (New England Biolabs) as per manufacturer’s instructions. Adapter ligation and clean-up were performed using NEBNext Quick Ligation Module (New England Biolabs) as per manufacturer’s instructions and the prepared library was loaded on the Nanopore SpotON Flow Cell (R9.4.1) and sequencing was performed over 60–70 h. Guppy basecaller (V4.0.11; https://nanoporetech.com/) was used to convert fast5 files to fastq files. Individual fastq files were concatenated and aligned with minimap2 (v2.16) [81] using the GETV_MM2021_ (MN849355) reference genome. Integrative genome viewer (IGV, v2.8.0) [82] was used to visualize sequence data and generate a consensus sequence. Across the 14 amplicons, the amplicon with the lowest read depth was approximately 13,000 and the highest was approximately 180,000.

A vial of GETV N544 (frozen in 1983) from the “Doherty Virus Collection” was thawed and virus grown in C6/36 cells. RNA was isolated from the culture supernatants using QIAamp Viral RNA Extraction kit (Qiagen, Chadstone, VIC, Australia). The RNA was used to transfect Vero cells using Xfect RNA Transfection Reagent (Takara Bio, Kusatsu, Japan) as per the manufacturer’s instructions. Cells were incubated for 3 days at 37 °C before the supernatant was transferred to C6/36 cells for 3 passages of 2–3 days each and supernatant was incubated with 10% PEG6000 (Sigma Aldrich, St. Louis, MO, USA) on a rotor overnight at 4 °C then centrifuged at ~134,000 rcf (Beckman Coulter, Brea, CA, USA) for 1 h at 4 °C. The pellet was resuspended in RAV1 lysis buffer from the Nucelospin RNA virus kit (Macherey-Nagel, Düren, Germany) and viral RNA was purified as per manufacturer’s instructions. Sequencing was conducted at the Australian Genome Research Facility (AGRF) where 890,405 paired-end reads (150 bp) were generated using the MiSeq platform (Illumina, San Diego, CA, USA). The Illumina bcl2fastq 2.20.0.422 pipeline was used to generate the sequence data. The paired-end reads were aligned to the GETV_MI-110-C2_ strain (LC079087.1) using STAR Aligner and the consensus sequence was obtained using Integrative Genomics Viewer (IGV).

### 4.4. Phylogenetic Tree Construction

An alignment was performed with GETV N544 and N554 consensus sequences and all available genome-length (and near-genome-length) GETV sequences in MEGA-X (Molecular Evolutionary Genetics Analysis 10, Penn State University, State College, PA, USA) using the ClustalW plugin with default parameters [83]. A phylogenetic tree was constructed as described [8] with the tree rooted using RRV_T48_, RRV_TT_, CHIKV Reunion Island isolate and southern elephant seal virus (SESV).

### 4.5. K-mer Mining and BLASTn Confirmation of High-Throughput Sequencing Data

High-throughput sequencing data sets deposited to the Sequence Read Archive (SRA) hosted by the National Center for Biotechnology Information, were screened using the BigQuery service offered by Google Cloud Computing Service. To identify SRA submissions that contained at least one “Getah virus” k-mer, the following Structured Query Language (SQL) command was used SELECT m.acc, m.sample_acc, m.biosample, m.sra_study, m.bioproject FROM nih-sra-datastore.sra.metadata as m, nih-sra-datastore.sra_tax_analysis_tool.tax_analysis as tax WHERE m.acc=tax.acc and NAME =‘Getah virus’ and total_count >1 ORDER BY m.bioproject, m.sra_study, m.biosample, m.sample_acc. SRA submissions identified as containing GETV k-mers were then separately queried for GETV sequences using Basic Local Alignment. The search tool BLASTn; (https://blast.ncbi.nlm.nih.gov/Blast.cgi?PROGRAM=blastn&PAGE_TYPE=BlastSearch) was used with the GETV_MM2021_ sequence (Genbank ID: MN849355) as the Query Sequence, and SRA selected for the Database and the Accession number entered. SRA submissions were deemed to contain GETV sequences if after BLASTn querying there were more than 4 reads greater than 40 bp in length with high pairwise nucleotide identity and Expected value scores (>90%; *E*-value: <5 × 10^−10^) to the reference GETV strain. For the RRV k-mer mining of SRA submissions, a similar pipeline was used; however, a higher k-mer count was employed (total count > 10). K-mer mining and BLASTn confirmation used the RRV reference (Genbank ID: MK028843; a 2009 clinical isolate). Representative alignments (to the GETV and RRV genomes) are shown in Appendix A.

### 4.6. Mice Infection and Vaccination

Female C57BL/6J mice (6–10-week-old) were purchased from the Animal Resources Centre (Canning Vale, WA, Australia). Mice were infected with 10^5^ or 10^6^ CCID_50_ of GETV_MM2021_ (Genbank ID; MN849355), 10^4^ CCID_50_ CHIKV Reunion Island isolate, LR2006OPY1 (Genbank ID; DQ443544) or 10^4^ CCID50 RRV_TT_ (Genbank ID; KY302801) subcutaneously (s.c.) into the top/side of each hind foot as described previously [8,54]. All virus preparations were mycoplasma free [84] as determined by MycoAlert Mycoplasma Detection Kit (Lonza, Basel, Switzerland). Serum viremia was determined by CCID_50_ assay, and foot swelling was determined using caliper height and width measurements, as described previously [8,54,85].

The JE/GETV formalin-inactivated vaccine (Nisseiken Co. Ltd., Tokyo, Japan) was used as described previously [8]. Mice were anesthetized with isoflurane and the vaccine was administered intramuscularly (i.m.) with the indicated dose split equally into both quadriceps muscles in 50 µL per muscle using an insulin syringe (Becton, Dickinson and Company, Franklin Lakes, NJ, USA).

### 4.7. ELISA and Neutralization Assays

Whole virus GETV_MM2021_ and RRV_TT_ antigens were purified from C6/36 tissue culture supernatants by polyethylene glycol (PEG) precipitation and sucrose cushion purification as described [8]. Alphavirus-specific IgG responses were determined by enzyme-linked immunosorbent assay (ELISA) using the whole alphavirus as antigen and the mean plus 3 standard deviations of serum from naïve mice as the endpoint as described [8]. The mean plus 3 standard deviations OD_405_ values ranged from 0.095 to 0.12, with the maximum OD_405_ ranging from 3 to 3.8. The limit of detection for convalescent sera was a 1 in 100 dilution (Figure 3A,B) and for sera from vaccinated mice was 1 in 30 (Figure 2C). 

Neutralization assay were performed as described [8]. Briefly, dilutions of heat-inactivated (56 °C, 30 min) mouse serum was incubated in 96 well plates with the indicated alphavirus for an hour at 37 °C. Vero cells (10^4^ per well) were then added to each well. After 5 days, cells were fixed and stained with formaldehyde and crystal violet, and 50% neutralizing titers interpolated from optical density (OD) versus serum dilution plots. Limit of detection for neutralization assays was 1 in 10.

### 4.8. Histology

Histology was undertaken as described previously [54]. Briefly, feet were fixed in 10% formalin, decalcified with EDTA, embedded in paraffin and sections stained with hematoxylin and eosin (H&E, Sigma-Aldrich, Darmstadt, Germany). Slides were scanned using Aperio AT Turbo (Aperio, Vista, CA, USA) and images extracted using Aperio ImageScope software v12.3.2.8013 (Leica Biosystems, Wetzlar, Germany).

### 4.9. Statistics

Statistical analyses were performed using IBM SPSS Statistics for Windows, Version 22.0 (IBM Corp., Armonk, NY, USA). Most of the data were non-parametric, with the difference in variances >4, so the Kolmogorov–Smirnov test or the Kruskal–Wallis test was used. Differences in foot swelling were analyzed using repeat measures ANOVA. 

## 5. Conclusions

The evidence that GETV circulates in Australia emerges not to be compelling. Early isolates purported to originate in Australia were the Malaysian GETV_MM2021_ isolate. Investigation of publicly available sequence read archives also found no evidence of GETV in Australian biosamples, whereas GETV sequences were readily identified in multiple Asian biosamples. Finally, the high level of cross-reactivity, cross-neutralization and cross-protection between RRV and GETV illustrated herein, suggests early serosurvey data may not be reliable.

## Figures and Tables

**Figure 1 pathogens-09-00848-f001:**
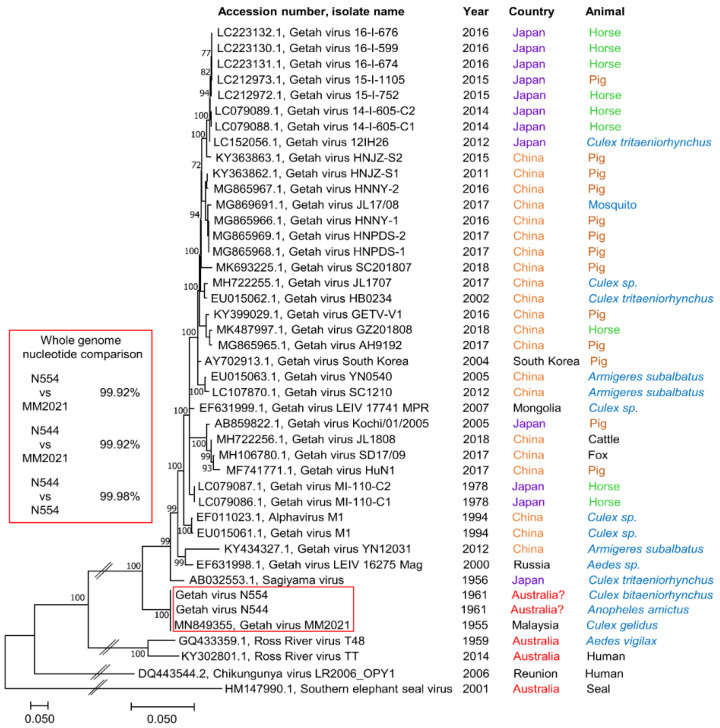
Phylogenetic tree of all full genome GETV nucleotide sequences. The phylogenic relationship between all submitted nucleotide sequences for genome-length GETV isolates, the year of isolation, the animal from which the isolate was obtained, and the country of origin. The red boxes highlight the near identity of N554, N544 and GETV_MM2021_; alignments are shown in Appendix A. The phylogenetic tree was constructed by using the Maximum Likelihood method and General Time Reversible model and 1000 bootstrap replicates. The percentage of trees in which the associated viruses clustered together is shown next to the branches. Branch lengths indicate the number of substitutions per site. The tree is rooted using southern elephant seal virus (SESV) [40] as an outgroup.

**Figure 2 pathogens-09-00848-f002:**
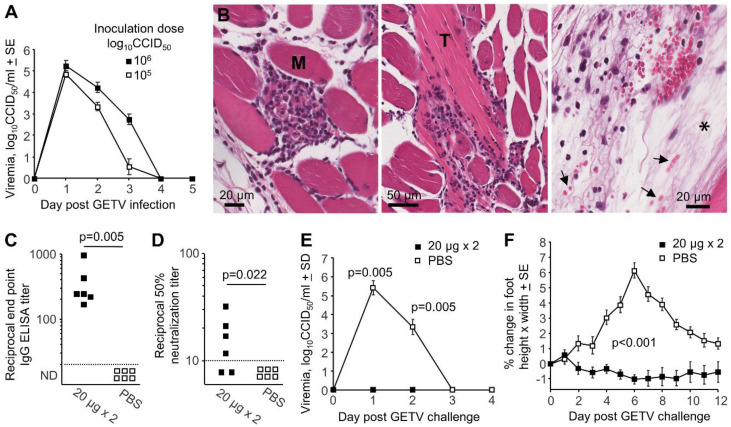
Adult wild-type mouse model of GETV infection and disease. (**A**) Viremia in adult female C57BL/6J mice after s.c. infection with the indicated dose of GETV_MM2021_ (*n* = 6 mice per group). (**B**) H&E staining of mice feet 6 days post-infection with 6 log_10_CCID_50_ of GETV_MM2021._ M—skeletal muscle. T—tendon. The high densities of purple nuclei illustrate inflammatory infiltrates. Arrows indicate examples of red blood cells clearly present outside blood vessels illustrating hemorrhage, with * indicating edema. (**C**) After 2 injections i.m. of 20 µg of JE/GETV vaccine, the serum endpoint ELISA titers (using the whole GETV as the ELISA antigen) are shown for 6 female C57BL/6J mice. As negative controls, 6 mice were mock vaccinated with PBS. (**D**) As for C but showing 50% endpoint neutralization titers for GETV_MM2021_. Statistics by Kruskal–Wallis test. (**E**) Mice in D were challenged s.c. with 5 log_10_CCID_50_ of GETV_MM2021_ and viremia determined. Statistics by Kolmogorov–Smirnov tests. (**F**) Percent change in foot height x width (foot swelling) for mice described in E. Statistics by repeat measure ANOVA for days 2–12 (*n* = 12 feet from 6 mice per group).

**Figure 3 pathogens-09-00848-f003:**
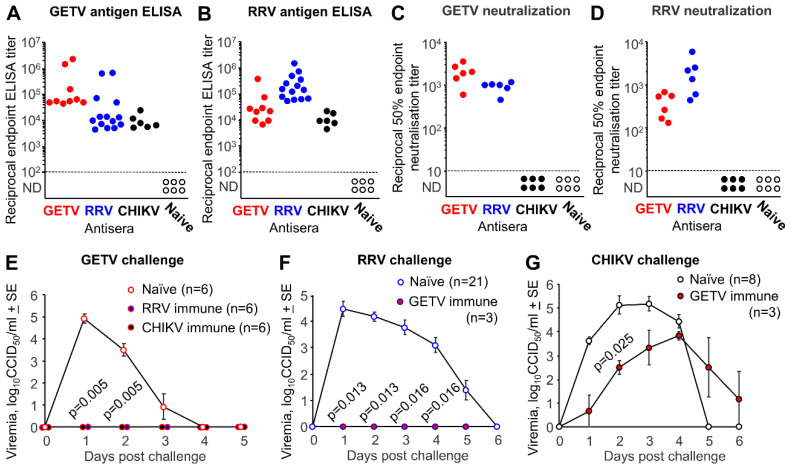
Cross-reaction and cross-protection between GETV and RRV. (**A**) Mice were infected with GETV, RRV or CHIKV and after 1 month, serum was used in a standard whole virus GETV antigen ELISA. Naive mice were used as negative controls. (**B**) The same serum samples described in A were used in an RRV antigen ELISA. (**C**) The same serum samples as in A were used in a neutralization assay using GETV_MM2021_. (**D**) The same serum samples as in A were used in a neutralization assay using RRV. (**E**) The indicated number of C57BL/6J mice were infected with RRV or CHIKV and after 3 weeks the mice were challenged with GETV and viremias determined. Statistics by Kolmogorov–Smirnov tests. (**F**) The indicated number of C57BL/6J mice were infected with GETV and after 3 weeks the mice were challenged with RRV and viremias determined. Statistics as for E. (**G**) The indicated number of C57BL/6J mice were infected with GETV and after 3 weeks the mice were challenged with CHIKV and viremias determined. Statistics as for E.

**Table 1 pathogens-09-00848-t001:** GETV k-mer and BLASTn positive SRA submissions. WGS—Whole Genome (DNA) Sequencing. RNA-Seq—RNA sequencing. Where multiple biosamples from the same Bioproject contained GETV k-mers and reads, the Accession number for only one biosample is listed. For instance, Bioproject PRJEB11005 contains 4 biosamples in the SRA submission where GETV k-mer sequences were found and confirmed by BLASTn, but only the Bioproject and one Accession is listed.

Accession	Bioproject	Country of Origin	Biosample
SRR1745767	PRJNA271540	China	RNA-Seq: Mosquito virome
SRR10014826	PRJNA561663	China	RNA-Seq: *Mus musculus* cells NIH3T3 cell line
SRR10728576	PRJNA596441	Sweden	RNA-Seq: *Danio rerio* **China** ecotype (zebrafish)
ERR1044994	PRJEB11005	China	WGS: *Homo sapiens* exome
ERR1341434	PRJEB12292	China	WGS: *Leuciscus waleckii* (fish)
ERR093004	PRJEB2869	China	WGS: *Homo sapiens* exome
ERR092416	PRJEB2869	China	WGS: *Homo sapiens* exome
SRR593462	PRJNA173904	Korea	WGS: *Homo sapiens* exome
SRR912432	PRJNA208608	China	RNA-Seq: *Zea mays* (maize)
SRR1020603	PRJNA224132	China	WGS: *Homo sapiens*
SRR1602104	PRJNA262923	China	WGS: *Homo sapiens* exome
SRR5171641	PRJNA360897	China	WGS: Mixed culture bioreactors
SRR5171639	PRJNA360897	China	WGS: Mixed culture bioreactors
SRR5481068	PRJNA384227	China	RNA-Seq: *Caenorhabditis elegans* (round worm)
SRR6032600	PRJNA406858	Taiwan	RNA-Seq: Activated sludge
SRR6260368	PRJNA416979	China	RNA-Seq: *Malus domestica* (apple)
SRR8733742	PRJNA421164	Hong Kong	RNA-Seq: *Mus musculus* (mouse)
SRR6668283	PRJNA432804	China	WGS: *Homo sapiens* blood
SRR6854500	PRJNA438879	China	miRNA-Seq: *Sus scrofa* (wild boar)
SRR6898032	PRJNA445618	China	RNA-seq: *Gymnocypris przewalskii* (fish)
SRR6917659	PRJNA445950	India	WGS: *Punica granatum* (pomegranate)
SRR8282611	PRJNA448784	China	RNA-Seq: *Halteria grandinella* (plankton)
SRR7141499	PRJNA470528	China	RNA-Seq: *Gallus Grandin Ella* (chicken) DF-1 cell line
SRR7250838	PRJNA472691	India	RNA-seq: *Glycine max* (soybean)
SRR7267677	PRJNA474550	China	RNA-Seq: Cultured Enterovirus
SRR7474260	PRJNA479403	China	RNA-Seq: *Escherichia coli*
SRR7811993	PRJNA490050	India	RNA-Seq: *Oryza sativa* (rice)
SRR8284531	PRJNA508497	China	RNA-Seq: *Pampus argenteus* liver (butterfish)
SRR8380233	PRJNA510861	China	WGS: *Cyprinus carpio* (Common carp)
SRR8365217	PRJNA511494	China	RNA-Seq: Sludge
SRR8731048	PRJNA526440	China	WGS: Activated sludge& biofilm
SRR8750958	PRJNA526443	Indonesia	WGS: Activated sludge& biofilm
SRR8920961	PRJNA532997	China	RNA-Seq: *Homo sapiens* lncRNA knock-down
SRR8983081	PRJNA540184	China	RNA-Seq: *Pampus minor* (pomfret, fish)
SRR9184436	PRJNA545960	China	RNA-Seq: *Mus musculus* C9orf72 cell line
SRR9184435	PRJNA545960	China	RNA-Seq: *Mus musculus* C9orf72 cell line
SRR9184437	PRJNA545960	China	RNA-Seq: *Mus musculus* C9orf72 cell line
SRR9265623	PRJNA548275	China	RNA-Seq: Human gastric cancer cell line
SRR9328883	PRJNA549716	China	RNA-Seq: *Serratia marcescens* (bacteria)
SRR9622339	PRJNA551957	China	ChIP-Seq WGS: *Glycine max*
SRR11511098	PRJNA556911	China	ChIP-Seq WGS: *Homo sapiens* 293T cells
SRR10054345	PRJNA559662	China	WGS: *Mycobacterium marinum* (bacteria)
SRR10134796	PRJNA566023	Japan	WGS: *Mus musculus*
SRR10161471	PRJNA573548	China	WGS: *Vibrio vulnificus* clinical isolate (bacteria)
SRR10586240	PRJNA593328	China	RNA-Seq: *Helianthus tuberosus* (artichoke)
SRR10727603	PRJNA596252	China	RNA-Seq: *Ovis aries* breed:Bashibay (sheep)
SRR10967930	PRJNA603230	China	RNA-Seq: *Escherichia coli* (bacteria)
SRR10992270	PRJNA604042	Taiwan	WGS: *Mus musculus*
SRR11188148	PRJNA608960	China	RNA-Seq: *Mus musculus*
SRR11341860	PRJNA610168	China	RNA-Seq: *Mus musculus*
SRR11290064	PRJNA611987	China	WGS: *Mus musculus*
SRR11657622	PRJNA624020	China	WGS: *Ovis aries* (sheep)
SRR11880368	PRJNA635796	China	RNA-Seq: *Chrysanthemum morifolium* (plant)
SRR11970763	PRJNA637815	China	WGS: *Escherichia coli* (bacteria)
SRR12396965	PRJNA645671	China	WGS: *Ovis aries* (sheep)
SRR354210	PRJNA76135	China	RNA-Seq: *Carthamus tinctorius* (safflower)

**Table 2 pathogens-09-00848-t002:** RRV k-mer and BLASTn positive SRA submissions. WGS—Whole Genome (DNA) Sequencing. RNA-Seq—RNA sequencing. Where multiple biosamples from the same Bioproject contained RRV k-mers and reads, the Accession number for only one biosample is listed. For instance, Bioproject PRJNA559742 contains 12 biosamples in the SRA submission where RRV k-mer sequences were found and confirmed by BLASTn, but only the Bioproject and one Accession is listed.

Accession	Bioproject	Country of Origin	Biosample
SRR5256949	PRJNA343688	Australia	RNA-Seq: Mosquito pools, Victoria
SRR12113269	PRJNA642916	Australia	RNA-Seq: Mixed mosquito species, Victoria
SRR11454617	PRJNA615690	Australia	Mosquito surveillance, Shoal Water Bay Defence Training Area, Queensland
SRR9948691	PRJNA559742	Australia	RNA-Seq: RRV spiked *Culex australicus*
SRR5572189	PRJNA386415	Australia	RNA-Seq: RRV infected *Ae. notoscriptus*
SRR8569108	PRJNA522026	Australia	RNA-Seq: RRV isolates
SRR11094162	PRJNA606985	Australia	WGS: Cephalosporin-resistant *Enterobacteriaceae*, Queensland
SRR12006518	PRJNA639216	Malaysia/Australia	WGS: *M. tuberculosis* Malaysia, submitted Charles Darwin University, Queensland
SRR8291079	PRJNA494517	China/US	WGS: *Plasmodium vivax* isolates China/Myanmar border, submitted by US lab

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
