# Peer review of "Sequencing of Historical Isolates, K-mer Mining and High Serological Cross-Reactivity with Ross River Virus Argue against the Presence of Getah Virus in Australia"

_pathogens, 2020, doi:10.3390/pathogens9100848_

Round 1

Reviewer 1 Report

This manuscript is well written and is suitable for publication in Pathogens with minor modification.

Line 406: Please provide the endpoint OD value.

Line 407: the authors stated limit of detection for vaccinated mice is 1:30. Which data uses this limit of detection?

Table 1 and Table 2: There are many GETV and RRV contamination in samples from various organism. Which gene of GETV and RRV have been detected in those samples?

Figure 1. If the virus GETV N544 and N554 were directly contaminated from GETVMM2021, it should be 100% identical to each other. However, the authors demonstrated that not 100% identical that identity being 99.92-99.98%. Is there any history of passaging GETVMM2021, N544 and N554 before sequencing? Please provide the passaging history of those stain if possible.

Author Response

Review #1

This manuscript is well written and is suitable for publication in Pathogens with minor modification.

Line 406: Please provide the endpoint OD value.

We have added these data to the Materials and Methods to illustrate the very comfortable signal to noise ratios; “The mean plus 3 standard deviations OD405 values ranged from 0.095 to 0.12, with the maximum OD405 ranging from 3 to 3.8”. 

Line 407: the authors stated limit of detection for vaccinated mice is 1:30. Which data uses this limit of detection?

This information has been added to the methods section; “Limit of detection for convalescent sera was a 1 in 100 dilution (Fig. 3A-B) and for sera from vaccinated mice was 1 in 30 (Fig. 2C).”

Table 1 and Table 2: There are many GETV and RRV contamination in samples from various organism. Which gene of GETV and RRV have been detected in those samples?

We have made some extensive changes to clarify our processes, in particular to clarify that k-mer mining identified high throughput sequencing data sets with possible GETV or RRV sequence reads.  This process does not identify where in the genome these “hits” have occurred, instead it identifies which SRA has potential GETV or RRV sequences using the whole genomes of the virus.  A separate confirmation process is then undertaken that aligned the high throughput sequencing data with the entire GETV or RRV genome using BLASTn.  When these aligned reads reached the indicated sequence length, homology and number, they were listed in Tables 1 and 2.  Thus the process identifies sequencing reads that would align to any part of the viral genome.  To exemplify the outcomes of the BLASTn search alignments for various SRAs we have added more alignments to Fig. S3 and have added an extra Fig. S4.  The reviewer’s suggested alignments actually provide additional data regarding the contamination, which we have added to the manuscript. 

Figure 1. If the virus GETV N544 and N554 were directly contaminated from GETVMM2021, it should be 100% identical to each other. However, the authors demonstrated that not 100% identical that identity being 99.92-99.98%. Is there any history of passaging GETVMM2021, N544 and N554 before sequencing? Please provide the passaging history of those stain if possible.

We have checked the files notes associated with the Doherty Virus Collection and have added the following sentence to line 103.  “The small differences are likely due to different passage histories with N554 frozen in 1961 after passage in mouse brains, N544 in 1983 after 5 passages in vitro and GETVMM2021 in 1986 after passage in mouse brains”. 

Reviewer 2 Report

The principal evidence that Getah virus circulates in Australia was derived from the two virus isolates the authors examined (along with serological studies). The authors recovered frozen Getah virus isolates that were first isolated over 50 years ago and completed whole genome sequencing. They probed open access sequence data via the Sequence Data Archuive  for evidence of Getah virus sequences in biosamples. They also explored the serological cross-reactivity of Getah and Ross River viruses. The sequencing studies found that these viruses were virtually identical to a 1955 isolate from Malaysia. Serological studies demonstrated sufficient cross-reactivity between GETV and RRV that previous exposure by either virus could not be distinguished in the mouse model. They also presented novel evidence (and approach) via k-mer mining that GETV sequences were not detected in biosamples from Australia, while those of RRV were. They were sufficiently conservative to state that more extensive samplings of  mosquito populations  in Australia may be warranted to support their findings that the evidence that GETV circulates in Australia is not compelling.

This is well written manuscript and I only have minor comments.

  • Line 145-146 Table 1. “For instance, Bioproject PRJEB11005 contains 4 biosamples in the SRA submission, with GETV k-mer sequences identified in all four, but only is listed below.” Make this last sentence that provides an example the same style as in Table 2.
  • Figure 2B (3rd image) – Does the accumulation of RBCs in the top right corner also reflect haemorrhage?

Author Response

Reviewer #2

The principal evidence that Getah virus circulates in Australia was derived from the two virus isolates the authors examined (along with serological studies). The authors recovered frozen Getah virus isolates that were first isolated over 50 years ago and completed whole genome sequencing. They probed open access sequence data via the Sequence Data Archuive  for evidence of Getah virus sequences in biosamples. They also explored the serological cross-reactivity of Getah and Ross River viruses. The sequencing studies found that these viruses were virtually identical to a 1955 isolate from Malaysia. Serological studies demonstrated sufficient cross-reactivity between GETV and RRV that previous exposure by either virus could not be distinguished in the mouse model. They also presented novel evidence (and approach) via k-mer mining that GETV sequences were not detected in biosamples from Australia, while those of RRV were. They were sufficiently conservative to state that more extensive samplings of  mosquito populations  in Australia may be warranted to support their findings that the evidence that GETV circulates in Australia is not compelling.

This is well written manuscript and I only have minor comments.

  • Line 145-146 Table 1. “For instance, Bioproject PRJEB11005 contains 4 biosamples in the SRA submission, with GETV k-mer sequences identified in all four, but only is listed below.” Make this last sentence that provides an example the same style as in Table 2.

The Table legends have been changed to be the same format for both tables, and to clarify what is shown; e.g. “Where multiple biosamples from the same Bioproject contained RRV k-mers and reads, the Accession number for only one biosamples is listed.  For instance, Bioproject PRJNA559742 contains 12 biosamples in the SRA submission where RRV k-mer sequences were found and validated by BLASTn searches, but only the Bioproject and one Accession is listed.

  • Figure 2B (3rdimage) – Does the accumulation of RBCs in the top right corner also reflect haemorrhage?

We have rephrased the figure legend to read “Arrows indicate examples of red blood cells clearly present outside blood vessels illustrating hemorrhage”.  The RBCs in the top right are likely to also be haemorrhage but the high density cluster could be a blood vessel (perhaps with vasculitis), so not quite so unequivocal.